# Cell Mechanics in Embryoid Bodies

**DOI:** 10.3390/cells9102270

**Published:** 2020-10-11

**Authors:** Kira Zeevaert, Mohamed H. Elsafi Mabrouk, Wolfgang Wagner, Roman Goetzke

**Affiliations:** 1Helmholtz-Institute for Biomedical Engineering, Stem Cell Biology and Cellular Engineering, RWTH Aachen University Medical School, 52074 Aachen, Germany; kzeevaert@ukaachen.de (K.Z.); mmabrouk@ukaachen.de (M.H.E.M.); 2Institute for Biomedical Engineering–Cell Biology, RWTH Aachen University Medical School, 52074 Aachen, Germany

**Keywords:** embryoid bodies, pluripotent stem cells, embryonic stem cells, induced pluripotent stem cells, cell mechanics, mechanical stimulation

## Abstract

Embryoid bodies (EBs) resemble self-organizing aggregates of pluripotent stem cells that recapitulate some aspects of early embryogenesis. Within few days, the cells undergo a transition from rather homogeneous epithelial-like pluripotent stem cell colonies into a three-dimensional organization of various cell types with multifaceted cell–cell interactions and lumen formation—a process associated with repetitive epithelial-mesenchymal transitions. In the last few years, culture methods have further evolved to better control EB size, growth, cellular composition, and organization—e.g., by the addition of morphogens or different extracellular matrix molecules. There is a growing perception that the mechanical properties, cell mechanics, and cell signaling during EB development are also influenced by physical cues to better guide lineage specification; substrate elasticity and topography are relevant, as well as shear stress and mechanical strain. Epithelial structures outside and inside EBs support the integrity of the cell aggregates and counteract mechanical stress. Furthermore, hydrogels can be used to better control the organization and lineage-specific differentiation of EBs. In this review, we summarize how EB formation is accompanied by a variety of biomechanical parameters that need to be considered for the directed and reproducible self-organization of early cell fate decisions.

## 1. Introduction

The first report of embryoid bodies (EBs) dates back to the mid-seventies, before the advent of embryonic stem cell (ESC) research [1]. The three-dimensional (3D) organization of EBs generated in vitro from mouse embryonic carcinoma cells was reminiscent of early embryonic structures, as it comprised cells from all three primary germ layers. The ability to form 3D structures that mimic early embryogenesis is inherent to all pluripotent stem cells, such as embryonic stem cells or induced pluripotent stem cells [2]. The latter can be reprogrammed from virtually any cell of the body and shares many important characteristics with ESCs [3,4]. Until now, different systems for EB generation from pluripotent stem cells have been developed and have provided important clues to unravel the sequence of early embryonic development.

In suspension culture, the aggregates of ESCs show a 3D organization of markers that resembles the initial segregation of extraembryonic and embryonic tissues: trophoectoderm, primitive endoderm, and early mesoderm [5]. Furthermore, the 3D aggregates of ESCs recapitulate developmental processes such as symmetry breaking, asymmetric gene expression, and approximate axis formation and elongation [2]. However, the in vitro culture systems for EB formation are rather simplistic and do not replicate the precise spatial and temporal organization during embryonic development [5].

Single-layered epithelial structures are the first differentiated cell types in the developing embryo [6]. Likewise, EBs are composed of an epithelial-like cell sheet on the surface which exhibits cell–cell adhesions that support self-organization [7]. Furthermore, the cavitation inside EBs facilitates the formation of a columnar epithelium, which is the precursor of the three definitive germ layers [8,9]. Little is known about the cellular response of EBs to mechanical stimuli. However, epithelial structures are extremely sensitive to mechanical challenges, and they can therefore mediate the cell mechanics in EBs [10].

There is clear evidence that mechanical forces interfere with the development of embryonic tissues [11]. Consequently, new culture systems for EB differentiation have been elaborated which allow the better control of cell mechanics in EBs. Such systems involve the control of EB size and cell–cell interaction; culture substrates with different extracellular matrix compositions; as well as mechanical stimuli, including compression, tensile, or shear forces in order to stimulate initial EB formation and 3D organization inside EBs. Nevertheless, the lack of standardized methods and quantitative measures hampers our understanding of how cell mechanics regulate the self-organization and lineage specification of pluripotent stem cells in 3D.

To address the effect of mechanical cues on the differentiation of pluripotent stem cells, standardized starting materials and culture conditions are urgently needed. In comparison to conventional 2D culture, the differentiation of 3D aggregates of pluripotent stem cells leads to increased induction into mesoderm and endoderm and the differential expression of genes that regulate developmental processes [12]. Thus, it is important to choose the best suited starting material and culture conditions to determine the effects of different mechanical stimuli on the cell fate decisions of aggregated pluripotent stem cells. In this review, we discuss the relevance of cell mechanics during EB differentiation and how mechanical stimulation affects the behavior and function of pluripotent stem cells in 3D aggregates.

## 2. Methods of Embryoid Body Formation

There are different approaches to generate EBs from pluripotent stem cells. The most commonly used method is simply to detach the pluripotent stem cells by enzymatic treatment and to then keep them in suspension culture, either in bacterial-grade petri-dishes or ultra-low attachment plates (Figure 1A). The stem cells will then spontaneously form non-adherent aggregates of differentiating cells [13,14,15]. However, the emerging EBs vary greatly in size and shape, which impacts the cell lineages that will emerge [2,16,17,18]. Embryoid bodies with a controlled size can be generated by the culture of pluripotent cells in “hanging drops” (Figure 1B). The sedimentation of the cells in a drop (50 µL), which hangs, e.g., from a petri-dish lid, results in the aggregation of the pluripotent stem cells by gravity and promotes efficient and more homogeneous EB formation [2,19,20]. The size of the EBs can be controlled by the number of cells per drop. Alternatively, the aggregation of pluripotent cells can be mediated by use of U- or V-shaped bottom well plates or plates with several thousand diamond-like grooves (Figure 1C,D) [21,22]. Here, the aggregation of a certain number of cells is driven by the gravity or centrifugation of the well plate, but particularly the latter might result in additional shear forces. Furthermore, the use of the bioprinting of cell-laden biological ink produces aggregates with a better control of spatial composition inside the resulting EBs (Figure 1E) [23]. Alternatively, it is also possible to use hydrogels in which the cells spontaneously form 3D aggregates (Figure 1F) [24,25]. All of the above-mentioned methods use a different approach to stimulate cell aggregation for EB generation, either natural or forced. This might impact the growth, homogeneity, and differentiation of the stem cell aggregates (Figure 1G,H) [26]. Thus, there is evidence that the different methods of EB formation affect cell mechanics inside EBs, while a systematic comparison of their relevance for cell fate decisions and multi-lineage differentiation still remains elusive.

## 3. Molecular Changes during Embryoid Body Formation

Another aspect of EBs that recapitulates early embryonic development is their temporal changes in gene expression and in the epigenetic landscape. For instance, similar genes are controlled in the early embryo and in the course of differentiation from the epithelial-like pluripotent cell colonies into the three germ layers [27,28,29,30]. Amongst the genes that are typically downregulated are key pluripotency factors such as POU Class 5 Homeobox 1 (*POU5F1;* OCT4) and *NANOG* [28,30]. The decline in the expression of the pluripotency gene network is gradual in EBs, and thus OCT4 can be detected up to 12 days after differentiation [5]. Furthermore, it has been shown that the OCT4-positive subpopulation of cells inside a human EB could be further classified into a subpopulation with neuroectodermal specification and a subpopulation that retains the co-expression of the pluripotency transcription factors OCT4 and REX1 [31]. The gradual decline of pluripotency-related genes upon EB differentiation is accompanied by an increasing expression of germ layer-specific genes, which represent early mesoderm (*TBXT*, *FOXA2, EOMES, CHRD*), endoderm (*GATA6*), and neuroectoderm (*SOX1, PAX6*) (Figure 2) [30]. Endoderm-like cells are first specified randomly at the core of the EB, where they lose the expression of OCT4 and overexpress endoderm-specific markers such as disabled-2 (*DAB2*), hepatocyte nuclear factor 4 (*HNF4*), and alpha-fetoprotein (*AFP*) [30]. In contrast, the gene expression of *TBXT* (Brachyury), indicating primitive streak formation [28], can be detected in cells which remain OCT4-positive. It has been shown that *TBXT* is expressed about 80 hours after the initiation of EB differentiation and thereupon reaches a peak after an additional 13 hours. Although the onset of the expression of *TBXT* varies between EBs, the spatiotemporal expression pattern after the onset shows little variation [32]. Following endoderm and mesoderm induction, the upregulation of *AFP* (endoderm marker) and NK2 Homeobox 5 (*NKX2.5*; cardiac differentiation marker) can be detected during the first week of EB differentiation [5,30].

The loss of pluripotency during EB differentiation is strongly related to the downregulation of the mechanotransducer Yes-associated protein (YAP), as shown during the differentiation of EBs generated from murine ESCs [33]. While YAP knockdown was suggested to lead to a loss of ES stemness, YAP-overexpressing cells showed high levels of alkaline phosphatase, a marker for pluripotency, and impaired neuronal differentiation capacity. Thus, YAP seems to play a critical role during the maintenance of ES cell pluripotency [33].

There is increasing variability in gene expression between individual EBs during differentiation [27]. This heterogeneity seems to be influenced by the starting conditions, culture methods, and differentiation conditions [34,35]. Furthermore, differentiation of EBs can be directed towards specific cell types by the activation/inhibition of different pathways. For instance, directed differentiation towards cardiomyocytes can be induced through the early activation of the Wnt pathway, which induces mesoderm differentiation. Accordingly, the 3D aggregation of human pluripotent stem cells was found to allow faster differentiation towards the cardiac lineage through the priming of mesodermal differentiation [36]. Moreover, EB size, the composition of different substrates, and mechanical cues can enhance the differentiation of EBs into specific lineages [37,38,39].

DNA methylation and chromatin accessibility seem to follow similar dynamics in EBs as in the early embryo. During mouse embryonic development, the global DNA methylation levels increase from 25% to approximately 75% in embryonic tissues and about 50% in extra-embryonic tissues between E4.5 and E5.5 [29]. Similarly, the onset of pluripotency exit in differentiating murine EBs is associated with a wave of de novo DNA methylation and a decrease in chromatin accessibility, which is reversed after germ layer specification [29]. Interestingly, ectoderm specification shows unique epigenetic dynamics compared to the other germ layers: the accessibility of the enhancer motifs of future ectodermal cells is conferred earlier than in cells that will undergo mesodermal or endodermal differentiation [29]. Furthermore, it has been shown that neurons which were differentiated through an EB assay accumulated atypical non-CG DNA methylation at the same rate as their in vivo counterparts [40]. However, during the late differentiation of EBs, the epigenetic signature of in vitro and in vivo cells starts to diverge [41].

## 4. Cell Mechanics and Lineage Specification in Embryoid Bodies

### 4.1. Modulation of Cell–Cell and Cell–Matrix Interaction during Embryoid Body Formation

Embryoid body formation relies on cell–cell interactions, which are primarily mediated by cadherins. Cadherins are Ca^2+^-dependent transmembrane adhesion receptors that trigger intracellular signaling by cytoplasmic catenin proteins—for example, β-catenin is linked to the Wnt pathway [42,43]. Undifferentiated ESCs highly express epithelial cadherin (E-cadherin), which is the primary molecular mediator of stem cell aggregation and compaction into EBs [12,44,45,46]. Consequently, the inhibition of E-cadherin-mediated adhesion prevents normal EB formation and subsequent germ layer differentiation [25]. The differential regulation of cadherins can influence EB differentiation and cell fate commitment. ESC lines with knockout for E-cadherin, which were rescued by the overexpression of E-cadherin, tended to form epithelium, whereas the overexpression of neural cadherin (N-cadherin) resulted in differentiation towards cartilage and neuroepithelium [45]. Therefore, the modulation of cadherin expression might be used to control EB formation and differentiation.

Apart from cadherin signaling, other strategies to control cell–cell interactions have been explored in stem cells. For instance, Notch signaling has been implicated to be involved in stem cell self-renewal and fate decision [7,47]. ESCs can express multiple Notch receptors that mediate the presentation of immobilized Notch receptor ligands from a surface such as a cell membrane in order to achieve optimal bioactivity [7]. Beckstead et al. have immobilized the Notch ligand Jagged-1 in an active conformation on a poly (2-hydroxyethyl methacrylate) (pHEMA) surface. The ligand that was bound to the surface stimulated the Notch/CBF-1 signaling pathway in epithelial stem cells and the cells upregulated both intermediate- and late stage differentiation markers [48]. Comparable methods of presenting Notch ligands to cells in EBs in order to stimulate differentiation would probably require modified biomaterials that are integrated within the cell aggregates or 3D hydrogels that include Notch ligands.

The epithelial cell sheet formation in embryonic structures is mainly driven by myosin II [49]. EBs with a deficiency of myosin II constantly shed cells due to decreased levels of E-cadherin and ß-catenin at cell–cell interfaces [50]. Thus, myosin II appears to play an important role in the formation of polarized endodermal structures, which form epithelial-like cell layers outside and inside EBs. In addition, the disruption of the cell–matrix interaction in embryoid bodies by blocking integrin α_v_β_3_, which recognizes Arg-Gly-Asp (RGD) motifs of extracellular matrix proteins, results in the deregulation of the endodermal layer, whereas the mesodermal and ectodermal layers are unchanged [25].

A common problem in EB formation is the low aggregation efficiency due to the insufficient control of cell–cell interaction. Therefore, de Bank et al. developed a method for forced cell aggregation by the biotinylation of murine ESCs (mESCs) and subsequent avidin cross-linking to produce homogeneous cell aggregates [42,51]. In comparison to conventional EBs, these engineered EBs formed more rapidly on the addition of avidin and appeared to be significantly denser and larger. The rapid establishment of cell–cell contacts in engineered EBs resulted in an enhanced osteogenic differentiation capacity compared to naturally formed EBs [51]. This is also reflected by a significantly higher expression of cell adhesion markers, such as cadherin-11, which is important for the osteogenic differentiation of mESCs, as well as the germ layer-specific proteins Brachyury (mesoderm) and Nestin (endoderm).

Not only cell–cell interactions within EBs but also the co-culture of EBs with differentiated cells can guide cellular differentiation and organization [52]. Petersen et al. engineered fibroblasts to express cadherins in order to adhere to preformed EBs of undifferentiated ESCs [52]. In these composite aggregates, the fibroblasts functioned as a discrete pole of cell differentiation. The further engineering of the fibroblasts to express *WNT3A* allowed the local upregulation of mesodermal marker genes in composite EBs. The cell–cell interaction between ESCs and fibroblasts, which express adhesion molecules and growth factors, allowed control over the timing and location of ESC differentiation in EBs.

### 4.2. Epithelial Cell to Mesenchymal Cell Transition (EMT) in Embryoid Bodies

The lineage specification in aggregates of pluripotent cells is not only influenced by the intercellular interaction, but particularly by intracellular adaptations, such as epithelial cell to mesenchymal cell transition (EMT). It resembles a switch from epithelial to mesenchymal cells by losing cell polarity and cell–cell adhesion, while gaining migratory and invasive properties. During embryo development, sequential rounds of EMT and vice versa are needed for the final differentiation of specialized cell types and the 3D organization of organs [53]. For instance, primitive streak formation after embryo implantation is determined by EMT events [53,54]. Thus, EBs recapitulate developmental EMT events and molecular and cellular changes that resemble primitive streak formation and mesendoderm [54].

During EB formation from human ESCs and iPSCs, a subpopulation of cells that reveal the first signals of EMT as shown by the upregulation of tyrosine-protein kinase-like 7 (PTK7) could be detected already after 24 hours [54]. PTK7 functions as a signaling switch for the Wnt pathway, which has been identified as the main pathway involved in EMT during primitive streak formation in different animal models [55,56]. Furthermore, cells that are positive for PTK7 overexpress several EMT-related genes, including alpha-smooth muscle actin (*α-SMA*), the Rho-related GTP binding protein *RHOJ*, and Ankyrin Repeat Domain 1 (*ANKRD1*) [54]. Accordingly, the modulators of the Wnt pathway affect the EMT process in EBs and subsequently impact on lineage specification during EB differentiation. For instance, the external addition of SPARC, which is upregulated in parietal endoderm at the onset of EMT, enhances EB differentiation towards cardiomyocytes [57]. In addition, it has been shown that EBs differentiated in rotatory suspension conditions demonstrate enhanced mesoderm differentiation compared to static cultured EBs via the modulation of the Wnt-ß-catenin pathway in combination with the upregulation of EMT-associated factors, such as *Snail* and *Brachyury* [58].

Failure to downregulate the E-cadherin expression is linked to impaired EMT [59]. The downregulation of E-cadherin during EMT is linked to an upregulation of the transcription repressor *Snail* [58]. As E-cadherin is downregulated, EBs start to upregulate the N-cadherin expression, which is referred to as “cadherin switch” [60]. The competition for β-catenin by both the Wnt pathway and cadherin-mediated cell–cell interactions impacts on cell–cell adhesion, the EMT process, and consequently lineage specification. For instance, methods of EB formation, which encourage strong cell–cell contacts, result in high E-cadherin levels accompanied by the downregulation of Wnt signaling, which leads to decreased cardiogenic differentiation [61,62]. This interplay shows the potential of optimizing the mechanobiological and cell mechanical niche to control EB differentiation.

### 4.3. Cellular Patterning and Germ Layer Specification during Embryoid Body Development

The in vitro culture of embryoid bodies allows detailed mechanistic insights into early differentiation events [7]. The outer layer of an embryoid body forms an epithelium, which resembles the embryonic primitive endoderm (Figure 3) [63]. The spontaneous formation of this layer of primitive endoderm is the first indication of the differentiation of EBs in suspension [7,9]. This process seems to be dependent on fibroblast growth factor (FGF) signaling, which is mediated by the PI-3 kinase pathway [64]. The primitive endoderm cells exhibit an epithelial morphology with tight cell–cell junctions and deposit a basement membrane that is rich in laminin and collagen IV, both of which are important for ectoderm differentiation and cavitation [7,46]. Constitutively active FGF signaling increases the assembly of laminin and collagen IV [46]. Furthermore, the basement membrane, which separates the primitive endoderm cells from the remaining undifferentiated cells within the EB, appears to support the survival of adjacent cells. In contrast, the inner cells that are not in direct contact with the basement membrane undergo caspase-dependent apoptosis, thereby contributing to the formation of cystic cavities within EBs [65,66,67]. Furthermore, the inner cells of the EB adopt an ectodermal fate and form a columnar epithelium resembling the epiblast [9]. Inner cells that are under the control of local Wnt-signaling undergo EMT, resulting in the development of mesoderm [9,56]. Thus, the maintenance of EBs in suspension culture results in the formation of various somatic cell types of endodermal, ectodermal, and mesodermal origin.

Alternatively, EBs can be transferred to tissue culture plastic coated with different extracellular matrix components, such as collagen. On such a substrate, EBs attach via the primitive endoderm, which might be related to the embedding of the blastocyst into the endometrium [9]. Subsequently, the primitive endoderm cells which surround the EBs and remain attached to the basement membrane differentiate into the extraembryonic visceral endoderm, which contributes to the visceral yolk sac of the mammalian embryo. RNA in situ hybridization suggests that BMP signaling is involved in the development of visceral endoderm [68]. In contrast, the primitive endoderm adjacent to the trophoectoderm differentiates into parietal endoderm, forming the parietal yolk sac [2,7,9]. Furthermore, both hematopoietic and cardiomyogenic progenitors emerge from mesodermal precursors within EBs in the form of yolk sac-like blood islands and spontaneously contractile foci, respectively [7,9,14]. The patterning of mesoderm is orchestrated by the influence of TGF-ß and Wnt signaling [56]. In addition to mesodermal cell types, such as endothelial cells and fibroblasts, also cells showing neurite extensions emerge from the EB. Thus, the expression of various protein markers of the endoderm (such as Foxa2, Sox17, GATA4/6, α-fetoprotein, and albumin), mesoderm (such as Brachyury-T, Msp1/2, Isl-1, α-actin, ζ-globin, and Runx2), and ectoderm (such as Sox1, Nestin, Pax6, GFAP, Olig2, neurofilament, and β–III Tubulin) validate the ability of EBs to generate cells of all three germ layers [7,69].

### 4.4. Molecular Pathways that Trigger the Cell-Mechanic Changes during EBs

As mentioned above, the 3D culture of pluripotent cells affects cell–cell contact and modulates developmental signaling pathways, including TGFβ/Activin/Nodal and BMP signaling, Notch, and Wnt/β-catenin signaling [12]. However, it is largely unclear what molecular mechanisms trigger these pathways. Comparative studies between EBs generated from human ESCs cultured as 2D monolayers and as 3D cell aggregates in microwells revealed significant differences in the expression of transcription factors that are important for mesendodermal differentiation [12]. For instance, the expression of SRY-box transcription factor 17 (*SOX17*), a marker for embryonic definitive endoderm, was detected one day earlier in EBs generated from micro-well-cultured ESCs as compared to EBs from 2D-cultured ESCs. Similarly, the expression of Brachyury, a transcription factor that controls mesendoderm induction, peaked at day 3 and 4 in EBs made from micro-well-cultured ESCs, whereas this peak did not occur in EBs made from ESCs cultured as a 2D monolayer [12].

The upregulation of the mesendoderm-specific genes in EBs generated with microwells is associated with the upregulation of genes responsive to Wnt signaling, such as the Wnt family members *WNT3A* and *WNT8A* and lymphoid enhancer binding factor 1 (*LEF1*) [12]. It has been shown that human ESCs cultured in 3D microwells exhibit a higher E-cadherin expression than cells on 2D substrates. This was accompanied by a downregulation of Wnt signaling, as evidenced by a lack of nuclear ß-catenin and the downregulation of Wnt target genes [61]. However, the EBs that were formed from ESCs in microwells demonstrated higher levels of Wnt signaling than the EBs generated from ESCs cultured on 2D substrates [61].

In addition, genes associated with EMT are highly upregulated in 3D ESC aggregates cultured in microwells as compared to ESCs grown in 2D [12]. The cells cultured in microwells revealed a reduced expression of markers associated with mesenchymal-like cells and EMT, such as fibronectin (*FN*) and N-cadherin (*CDH2*). These results suggest that ESCs cultured in 3D microwells exhibit a more epithelial phenotype, possibly due to the increased cell–cell contact to neighboring cells [12]. This finding has further been substantiated by the detection of increased levels of E-cadherin in ESCs cultured in microwells as compared to 2D-cultured ESCs on plastic. On the other hand, genes related to TGFβ/Activin/Nodal signaling, such as *NODAL* and growth differentiation factor 11 (*GDF11*), exhibited a reduced expression in ESCs in microwells. These results may indicate that there is less signaling through the TGFβ/Activin/Nodal pathway when cells are cultured in 3D microwells.

## 5. Impact of Size and Mechanical Stimulation on Differentiation of EBs

### 5.1. The Size of Stem Cell Aggregates Modulates Their Differentiation Potential

The differentiation of EBs into specific lineages is tightly connected to the size. For example, if EBs are formed by the detachment of pluripotent stem cell colonies, the size of the 2D colonies determines the number of cells of the resulting aggregate (Table 1) [70]. Alternatively, the EB size, as well as local cell density within the EB, can be controlled by cell printing technology [71]. A comparison of murine EBs of different sizes revealed that an intermediate size of 100–300 µm leads to higher viability, proliferation, and differentiation potential of the enclosed cells. Furthermore, hydrophobic surfaces such as polydimethylsiloxane (PDMS) were used to generate viable EBs of a homogeneous size that revealed an improved differentiation towards hematopoietic lineages as compared to EBs generated in ultra-low attachment plates [72]. Additionally, it was shown that the size of EBs generated from microwells controls the differential expression of different Wnt target genes; the *WNT5a* expression was highly upregulated, whereas the *WNT11* expression was absent in small EBs (150 µm in diameter) and vice versa in large EBs (450 µm in diameter) [34]. Due to the differential expression of *WNT5a* and *WNT11*, small EBs tended to differentiate into endothelial cells, whereas large EBs were more prone to differentiate into cardiac cell types.

Murine EBs of different sizes demonstrated large variations in their global gene expression profiles: smaller aggregates with a diameter of 100 µm showed increased ectoderm (*HES1*) marker expression, while aggregates with a diameter of 500 µm demonstrated an increased expression of endoderm (*AFP*) and mesoderm (*GATA1*, *Nkx2.5*, *Foxf1*)-associated markers. Intermediate-sized EBs with a diameter of 300 µm equally supported mesoderm, ectoderm, and endoderm-specific marker expression [73]. Furthermore, Bauwens et al. analyzed the gene and protein expression of size-controlled human EBs. A decreasing colony size was associated with an increasing *GATA6* to *PAX6* (endoderm to ectoderm) ratio. Their results underlined that the differentiation of ESCs in 3D depends on a critical size of the aggregate, and that the initial size is a crucial factor which controls the late-stage differentiation [70]. Consequently, size control enables the better-defined differentiation of EBs.

### 5.2. Effect of Shear Stress on Embryoid Body Differentiation

Shear stress, a mechanical force resulting from the dynamic or continuous flow of a fluid, has a strong impact on stem cell differentiation and embryonic development (Table 1) [74]. For instance, it has been shown that murine ESCs subjected to shear stress show an increased expression of markers for vascular endothelium [75]. Furthermore, shear stress may enhance the viability and differentiation of EBs. A microfluidics system that generates shear stress by rotary force with a rotary speed of 40 rpm resulted in a 20-fold enhancement of the number of cells incorporated in murine EBs as compared to EBs that were formed by hanging drop or static methods [76]. In addition, the endoderm-specific gene expression and cyst formation were enhanced in rotary EBs, while the pluripotent differentiation capacity was retained [76].

The effect of shear stress on EB differentiation can be studied by diverse microfluidics systems. For instance, Guven et al. have developed a microfluidic chip system that allows the application of laminar flow on EBs and simultaneously collects hormones from the supernatant [77]. In this system, the laminar flow mediates estradiol and progesterone secretion in murine steroidogenic stem cell-derived EBs at physiologically relevant concentrations for up to 21 days [77]. Fung et al. have created a microfluidic platform consisting of a Y-shaped channel device with two inlets which are flooded with two different culture media. Murine EBs were placed across both streams, and by independently culturing each half of the EB with different media, cell differentiation was induced in one half of the EB, while the other half remained unchanged [78]. In principle, such a device would also allow the treatment of EBs with two different flow rates to determine the effect of shear stress on different areas of the cell aggregate.

The preconditioning of ESCs with shear stress prior to 3D cell aggregation affects the molecular phenotype of differentiating EBs. A comparative study investigated the effect of shear stress (5 dyn/cm^2^) and static conditions on ESC monolayers 48 hours before the initiation of EB differentiation [79]. Initially, the cells cultured under shear stress and static conditions retained a pluripotent phenotype and exhibited similar morphologies when they were aggregated into EBs. However, after one week 70% of the EBs from the shear stress condition expressed significantly higher levels of endothelial markers (Flk1, E-cadherin, and PECAM) than EBs from statically cultured ESCs. Additionally, the fluid shear stress seemed to impact on the organization of endothelial cells within the EBs in the form of a centrally localized region of VE-cadherin-positive cells.

Furthermore, shear stress supports the growth and maturation of EBs. The cellular organization, morphology, and gene expression profiles of EBs were compared after stimulation under static and hydrodynamic conditions with a fluid shear stress ranging from 0.7 to 2.5 dyn/cm^2^ [80]. A slower rotary speed correlated with larger EBs, ranging from a cross sectional area of 200,000 µm^2^ at 20 to 25 rpm to 15,000 µm^2^ at 60 rpm. After one week, large EBs at a low rotary speed consisted of a necrotic, hollow core surrounded by multiple layers of cells, while cystic structures and cellular variability were reduced at a higher rotary speed. An analysis of the gene expression profiles revealed a higher expression of pluripotency markers (Oct4 and Nanog) by EBs cultured in the static condition, indicating an increase in the differentiation potential of EBs in hydrodynamic culture. Accordingly, the germ layer-specific marker gene expression was enhanced in rotary orbital suspension conditions. Overall, stem cell aggregation and maturation were more homogeneous and more efficient if the EBs were affected by shear stress.

### 5.3. Impact of Mechanical Strain on EBs

A major challenge for the analysis of mechanical effects on EBs is to provide a defined and quantifiable stimulus (Table 1) [81]. Defined strain can be initiated by magnetism. For instance, RGD (Arginine-Glycine-Aspartic Acid)-conjugated paramagnetic beads were incorporated into the interior of murine EBs and an array of neodymium magnets with different magnetic field strengths underneath the EBs facilitated the continuous attraction of the EBs to the magnets. After stimulation in a 0.2 Tesla magnetic field for one hour, cells within the EBs started to activate Protein Kinase A (PKA) expression and increased levels of phosphorylated extracellular signal-regulated kinase 1/2 (pERK1/2), both of which are associated with integrin activation. In contrast, higher (0.4 Tesla) or lower (0.128 Tesla) magnetic field strengths did not have a stimulating effect on the EBs. Furthermore, treatment with BMP4 in combination with one hour of magnetic attraction with a field strength of 0.2 Tesla on day three of differentiation enhanced cardiomyogenesis, while magnetic nanoparticle internalization itself had no influence on the differentiation of EBs [82].

Cyclic stretch and compression of magnetically labelled EBs using two opposing attractor microtips led to similar results [39]. Here, the EBs were either stretched once up to 50% of their original size (stretched condition) or further stretched twice a day for 2 hours at a frequency of 1 Hz and amplitude of 10% of the EB size for three more consecutive days (cyclic condition). In contrast to hanging drop EBs, stretching stimulation initiated stem cell differentiation towards mesodermal lineages; the expression of Nkx2.5, Sox17, Gata4, and Gata6 (cardiac mesoderm markers) was increased in the stretched condition and even more enhanced in the cyclic condition. Alternative experiments using only one microtip instead of two did not change gene expression because mechanical strain was only present at a limited region of the EB, whereas the majority of the cells was exposed to negligible stress [39]. Consequently, magnetic devices can be used for the application of defined stretching or compression forces to 3D cell aggregates.

Alternatively, strain can be applied by mechanical stretching. The use of a Flexercell Strain Unit in combination with flexible-bottomed culture plates allowed 10% elongation of EBs for up to two hours [83]. As a result, mechanical strain increased the vascularization of EBs during differentiation, as demonstrated by a significant increase in the pro-angiogenic growth factor expression, including vascular endothelial growth factor (VEGF), platelet-derived growth factor (PDGF-BB), and fibroblast growth factor (FGF-2) [83]. The stimulating effect of 10% elongation of EBs on cardiomyogenesis has been further indicated by an increased capillary area in EBs and a significant increase in spontaneously contracting cardiac foci [84]. Different studies showed that the mechanical strain-induced cardiovascular differentiation in EBs is dependent on the increased generation of reactive oxygen species (ROS), followed by an upregulation of different mitogen-activated protein kinases (MAPKs), which are necessary for cardiomyogenesis [83,84].

### 5.4. Hydrogels for EB Differentiation

Recent studies have shown that stem cells preferentially differentiate towards distinct cell types on substrates that provide an artificial 3D environment (Table 1) [85,86,87]. Characteristics such as elasticity, degradability, and bioactivity can be easily modulated by the use of hydrogels composed of synthetic components such as polyethylene glycol (PEG) [88,89]. Continuous 3D projection printing was used to generate concave hydrogel microstructures made of photo-cross-linkable PEG diacrylate (DA) to allow spheroid growth and long-term maintenance without the need to transfer the spheroids. In general, the EBs generated from iPSCs in these structures demonstrated a narrower size distribution (155 ± 17 µm) compared to flat microwells (129 ± 48 µm) of the same diameter. Furthermore, single EBs housed in the concave hydrogel demonstrated an increase in the expression of germ layer-specific markers (Brachyury, SOX-17, and SOX-1), as well as cavity formation after 10 days [90].

Hydrogels can stimulate EB polarity. Qi et al. fabricated hybrid 3D microengineered hydrogels consisting of two neighboring hydrogels, photo-cross-linkable gelatin methacrylate (GelMA) and PEG [91]. Both hydrogels have a different bioactivity. Due to the enhanced presence of gelatin macromolecules, GelMA (3 wt%) facilitated the distinct and robust sprouting of mouse ESC aggregates, whereas the inert PEG hydrogel (10 wt%) inhibited cell proliferation while the EBs stayed viable [91]. Furthermore, EBs embedded at the interface of both hydrogels revealed developmental polarities with patterned vasculogenic differentiation due to cross-talk with the different microenvironments. It could be shown that, in contrast to PEG, GelMA stimulates the expression of pro-vasculogenic differentiation factors, including platelet endothelial cell adhesion molecule (PECAM-1) and angiopoietin receptor Tie-2 [91].

Furthermore, the effect of hydrogels on stem cell differentiation in a 3D microenvironment was addressed by use of microwell arrays for uniform murine EB generation (150 µm and 450 µm EBs) prior to encapsulation in PEG hydrogels with and without conjugated RGD [92]. Generally, increasing EB size was coupled with a reduced amount of total EBs in PEG hydrogel. Moreover, EBs in PEG hydrogel aggregated, while in RGD-PEG, EBs stayed separated. This is probably due to the nature of the RGD peptides, which provided cell–matrix interaction and thereby stabilized EBs. Furthermore, pristine PEG hydrogels allowed size-controlled EBs to interact with each other, which promoted larger aggregates with enhanced cardiogenic differentiation, as shown by the increased beating signals [92]. In contrast, there was a much lower frequency of EBs in RGD-PEG hydrogel, which revealed spontaneous contractions. On the other hand, the RGD peptides inside PEG hydrogel promoted endothelial cell differentiation in size-controlled EBs, as shown by the enhanced sprouting and upregulation of vasculogenic markers (CD31, VE-Cadherin).

**Table 1 cells-09-02270-t001:** Studies on the effect of size and mechanical stimulation on EB differentiation.

	Species	Cell Type	Mechanical Stimulus	Parameters Tested	Duration	Readout	Ref.
Substrate Elasticity	Human	ESC	Rigidity of PDMS micropost arrays	1.92–1218.4 kPa	24 hours	Traction force measurements; pluripotency and cytoskeletal changes (IF)	[93]
	Human	iPSC	Elastomer pillars of different heights on a rigid substrate	3–168 kPa	Up to 8 days	Morphology; viability; proliferation; pluripotency (IF); cardiac differentiation (IF, FACS, Ca^2+^ assays, electrical stimulation)	[85]
Substrate Topography	Human	iPSC	Polyimide substrate with submicrometer groove-ridge structure	340 nm, 650 nm, 1400 nm periodicity, 200 nm depth; compared to unstructured	3 days	Morphology; pluripotency (IF, qRT-PCR); YAP/TAZ expression (WB), gene expression profiles	[94]
Colony Size	Human	ESC, EB	Size of initial 2D colony and EB size via micropatterning	200, 400, 800 µm	4–22 days	Morphology; pluripotency (FACS) and germ layer differentiation (qRT-PCR, IF)	[70]
	Mouse	ESC, EB	Size of initial 2D colony and cell density via laser direct-write cell printing	Colony size: 200–3000 µm; cell density: <25,000 cells, 25,000–125,000 cells, >125,000 cells	3–8 days	Morphology	[71]
	Mouse, human	ESC, EB	Size of EBs; substrate hydrophobicity	Colony size: <100 µm, 100–300 µm, >300 µm; surface chemical properties: agarose, PEG, pHEMA, PDMS, TCP, LAC	4–20 days	Morphology; viability; proliferation; (germ layer) differentiation potential (qRT-PCR, FACS)	[72]
	Mouse	ESC, EB	Size of EBs via adhesive stencils with different diameter	100–500 µm diameter	20 days	Germ layer differentiation (IF, qRT-PCR)	[73]
Shear Stress	Mouse	ESC	Laminar shear stress	1.5–10 dyn/cm^2^	3 days	Cell density; cell cycle (FACS, ELISA); endothelial differentiation (IF, WB, qRT-PCR)	[75]
	Mouse	ESC, EB	Shear stress in rotary suspension culture	Comparison of static and rotary suspension culture (25–55 rpm)	12 hours to 7 days	Morphology; viability; proliferation; cyst formation; germ layer formation (qRT-PCR)	[76]
	Mouse	ESC, EB	Y-channel microfluidic system with two different media	Laminar flow 50–200 µL/min	5 days	Differentiation potential (WB, IF)	[78]
	Mouse	EB	Microfluidic chip system with continuous laminar flow and shear stress	Comparison of static culture and laminar flow 2 µL/min	21 days	Viability; proliferation; steroidogenic differentiation (hormone release, IF, ELISA)	[77]
	Mouse	ECS, EB	Comparison of static conditions and shear stress as pre-condition	0–5 dyn/cm^2^	48 hours before EB formation; up to 10 days of EB culture	Morphology; pluripotency and endothelial differentiation (IF, qRT-PCR, FACS) cellular organization	[79]
	Mouse	ESC, EB	Shear stress in rotary orbital suspension culture	0.7–2.5 dyn/cm^2^; 20–60 rpm	EB culture for 7 days; up to day 12 of differentiation	Morphology (IF); pluripotency and germ layer differentiation (qRT-PCR, FACS); global gene expression (PCR array analysis)	[80]
Mechanical Strain	Mouse	ESC, EB	Mechanical strain by short-term magnetization via incorporated RGD-conjugated paramagnetic beads	Stimulation pulses using short-term magnetization; 0.128–0.4 Tesla	1 hour stimulation for up to 7 days	Morphology; viability; protein expression and cardiomyogenesis (β1 integrin inhibition, FACS, WB, IF)	[82]
	Mouse	ESC, EB	Mechanical strain by (cyclic) stretching and compression between two microtips with a magnetic tissue stretcher	Stretching amplitude of 50% of original size (+ cyclic stretching 1 Hz, 10% amplitude, twice daily for 2 hours)	3 days; with three additional days of cyclic stretching	Morphology; viability; proliferation; pluripotency and germ layer differentiation (qRT-PCR, IF)	[39]
	Mouse	ESC, EB	Mechanical strain by stretching device (Flexercell Strain Unit)	10% elongation of undifferentiated EBs	2 hours stretching on day 4 of EB generation	Monitoring of intracellular [Ca^2+^]_i_; Expression of angiogenesis guidance molecules; Expression of pro-angiogenic growth factors; ROS generation	[83]
	Mouse	ESC, EB	Mechanical strain by stretching device (Flexercell Strain Unit)	5%, 10%, or 20% elongation of undifferentiated EBs	2 hours stretching on day 4 of EB generation	Staining of capillary-like structures; counting of beating bodies; staining of sarcomeric α-actinin; upregulation of NADPH oxidase subunits; ROS generation; inhibition of mechanical-strain stimulated MAPKs	[84]
3D Culture in Hydrogel	Human	ESC, EB	Agarose 3D culture system	Comparison of agarose 3D culture and suspension and hanging droplet culture	7 days up to 8 weeks	In vivo teratoma assay; morphology; germ layer differentiation (IF)	[95]
	Human	ESC	Hydrogel based material for switching between alginate and collagen via ionic de-crosslinking	Change hydrogel composition and in matrix elasticity from 21.37 ± 5.37 kPa (alginate) to 4.87 ± 1.64 kPa (collagen)	21 days	Viability; proliferation; pluripotency and germ layer differentiation (qRT-PCR)	[96]
	Human	ESC, EB	Dextran-acrylate and PEG hydrogel +/- RGD and VEGF	Comparison of dextran-acrylate and PEG hydrogel +/- RGD and VEGF	10 days	Viability; vascular differentiation (FACS, IF, qRT-PCR)	[88]
	Human	ESC, EB	Biodegradable polymer scaffolds (50:50 PLGA:PLLA)	Medium supplemented with different growth factors	14 days in vitro; 14 days in vivo	Proliferation; germ layer differentiation (qRT-PCR); transplantation into SCID mice (IF)	[89]
	Human	iPSC, EB	Concave PEG hydrogel microstructures via 3D projection printing; initial cell number	Comparison between concave and flat gels; low (250,000/mL) and high (750,000/mL) cell density	Up to 10 days	Morphology; culture duration; pluripotency and germ layer differentiation (IF); cyst formation	[90]
	Mouse	ESC, EB	Hybrid hydrogels (GelMA, PEG) with varying matrix elasticity	Analysis of a hybrid GelMA (3 wt%)/PEG (10 wt%) hydrogel	Up to 7 days	Vasculogenic and cardiogenic differentiation (qRT-PCR, IF)	[91]
	Mouse	ESC, EB	PEG hydrogel with and without RGD	Comparison of different PEG gels; 150 µm EBs and 450 µm EBs	Up to 15 days	Morphology; endothelial and cardiac differentiation (contraction behavior, IF, qRT-PCR)	[92]

This table exemplarily summarizes studies that investigated mechanical stimuli applied to EBs as mentioned in the text. Please note that this selection is incomplete, and we apologize for not being able to highlight all the relevant studies in this review. ESC = embryonic stem cells; iPSC = induced pluripotent stem cells; EB = embryoid body; RGD = Arg-Gly-Asp (RGD) motifs; VEGF = vascular endothelial growth factor; TCP = tissue culture plastic; LAC = ultra-low-attachment culture; PDMS = polydimethylsiloxane; PEG = polyethylene glycol; GelMA = gelatin-methacrylol; PLGA = polylactid-co-glycolid; PLLA = polylactide; pHEMA = poly(2-hydroxyethyl methacrylate); IF = immunofluorescence; FACS = fluorescence-activated cell sorting; WB = Western Blot; ROS = reactive oxygen species; MAPKs = mitogen-activated protein kinases.

## 6. Conclusions

The transition from an epithelial-like flat colony of pluripotent stem cells into a complex EB with multiple cell types is a fascinating process that involves many mechanical aspects. The self-organization of EBs supports multi-lineage differentiation and enables the detailed in vivo analysis of relevant cell-intrinsic and cell-extrinsic parameters. Some of these aspects are also recapitulated by teratoma formation upon the in vivo injection of pluripotent cells, but this assay is based on controversial animal experiments. Furthermore, the analysis of EBs is faster, more cost-effective, and enables easier quantification. The methods for EB generation and subsequent histological and molecular analysis have gone through tremendous refinement, taking into account the complex interplay between physical forces and the mechanical properties of cell aggregates. Especially the epithelial structures outside and inside EBs counteract such mechanical cues and stabilize their spherical structure. The still relatively small number of studies on cell mechanics in 3D cell aggregates points towards the need for a more holistic understanding of the importance of mechanical stimulation for cell fate decisions. Such approaches can be used for spatially controlled stem cell differentiation, which may provide new perspectives for tissue engineering and regenerative medicine.

## Figures and Tables

**Figure 1 cells-09-02270-f001:**
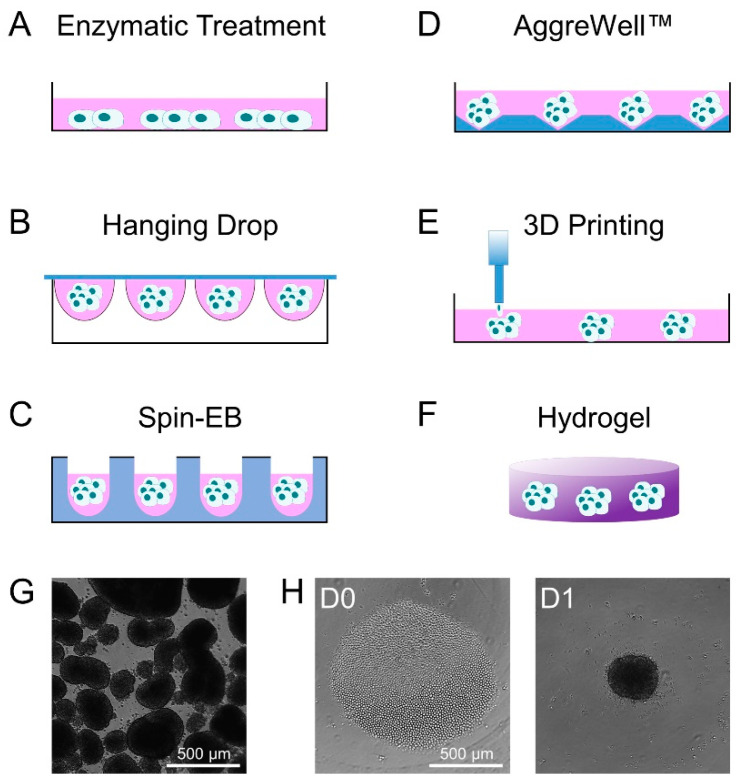
Schematic representation of commonly used methods for the formation of embryoid bodies. (**A**) EBs are commonly produced from monolayer cultures of pluripotent stem cells using enzymatic treatment to fragment and detach cell colonies from the culture well. (**B**) Seeding single pluripotent stem cells in hanging drops facilitates the generation of homogeneous EBs. Alternatively, single cells can be spun down in (**C**) U-shaped bottom wells or (**D**) AggreWells^TM^ to facilitate homogeneous cell aggregation. (**E**) Bioprinting using cell-laden biological ink allows better control over the composition of EBs. Finally, (**F**) hydrogels provide an elastic 3D environment which allows the spontaneous formation and differentiation of EBs. (**G**) Exemplary phase contrast images of EBs produced by enzymatic treatment and (**H**) EBs produced by a Spin-EB assay starting with a specific cell number (D0) to generate aggregates that are more uniform in size (D1).

**Figure 2 cells-09-02270-f002:**
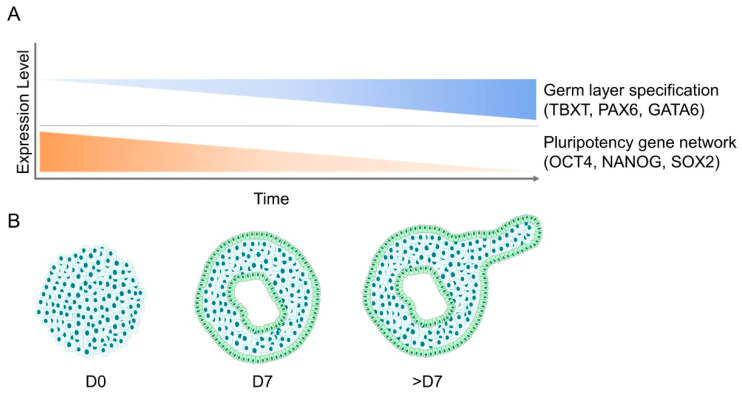
Gene expression and morphological changes during the course of EB development. (**A**) Gene expression patterns of embryoid bodies show simultaneous upregulation of germ layer-specific markers (Brachyury, PAX6, GATA6) and downregulation of pluripotency-related genes (OCT4, NANOG, SOX2) over the course of their differentiation. (**B**) Morphologically, EBs transition from a dense mass of cells into fluid-filled cavitated structures that can later develop additional appendages.

**Figure 3 cells-09-02270-f003:**
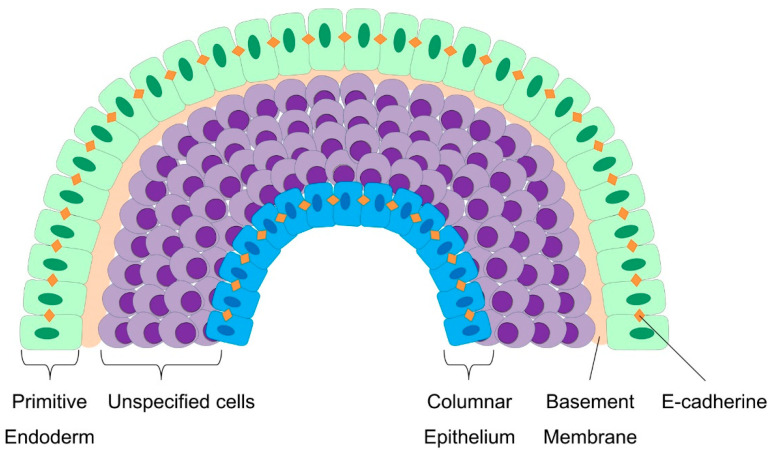
Cellular patterning during embryoid body formation. Around day 2, cells on the surface of the EB form a layer of primitive endoderm which exhibits an epithelial morphology with an increased expression of E-cadherin and a sensitive response to FGF signaling. These cells deposit a basement membrane which is rich in laminin and collagen IV. Cells adjacent to the basement membrane receive survival signals, whereas cells without contact undergo caspase-dependent apoptosis. Cavitation inside the EB forms a yolk sac-like structure and results in the formation of a columnar epithelium. Cells between the primitive endoderm and the columnar epithelium are highly responsive to different signaling pathways, including Wnt, Nodal, and BMP signaling.

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
