# Peer review of "Cell Mechanics in Embryoid Bodies"

_cells, 2020, doi:10.3390/cells9102270_

Round 1

Reviewer 1 Report

In this review manuscript titled: "Cell mechanics in embryoid bodies", the authors provide a comprehensive review on the research involving EB formation. The text is well-written and easy to follow. The subject is well-covered and provides information on the various aspects of EB formation, such as methods, cell processes, gene expression, and the infuence of various parameters of the culture environment, such as mechanical properties of the substrate, sheer stess, etc. on EB quality. This review would be very useful and informative to scientists who work with pluripotent stem cells and conduct in vitro differentiations as part of their analyses. Since I do not see any problems with the manuscript, I would like to recommend it for publication.

Author Response

We thank the reviewer for his encouraging words.

Reviewer 2 Report

 This review is summarizing data about the processes that are going on when cells undergo transition from the rather homogeneous epithelial-like pluripotent stem cell colonies into a three-dimensional organization of various cell types – a process associated with epithelial-mesenchymal transitions. The parts of review article - Molecular changes during embryoid body formation, as well as changes in EMT process are informative, well written with many concrete data and citations. However, the paragraph 5 needs some corrections.

Remarks.

  1. The paragraph No 5 needs some revision and extension. 5.1 part is about the impact of size of stem cell aggregates to the internal processes, however the Table 1 includes both, the effects of size and mechanical stimulation, which complicate following the data. These two effects can be separated into two tables. Table with mechanical stress can be added to the 5.3 part.
  2. The parts 5.3, as well as 5.4, are written quite superficially and could be extended.

Author Response

Response to Reviewer #2:

This review is summarizing data about the processes that are going on when cells undergo transition from the rather homogeneous epithelial-like pluripotent stem cell colonies into a three-dimensional organization of various cell types – a process associated with epithelial-mesenchymal transitions. The parts of review article - Molecular changes during embryoid body formation, as well as changes in EMT process are informative, well written with many concrete data and citations.

We thank the reviewer for the critical input.

 […] However, the paragraph 5 needs some corrections.

Remarks.

  1. The paragraph No 5 needs some revision and extension. 5.1 part is about the impact of size of stem cell aggregates to the internal processes, however the Table 1 includes both, the effects of size and mechanical stimulation, which complicate following the data. These two effects can be separated into two tables. Table with mechanical stress can be added to the 5.3 part.

We have moved Table 1 to the end of section 5.4 (Page 11, line 457). We appreciate the reviewer´s suggestion to split the table into two, however, we feel that the table in its current form provides a more comprehensive overview on the different parameters that have been applied to influence embryoid bodies.

  1. The parts 5.3, as well as 5.4, are written quite superficially and could be extended.

We have better specified the findings described in section 5.3 and added further studies to this section (Page 10, line 409-419) and to Table 1. Furthermore, we have also described the findings in section 5.4 in more detail. 

We thank both reviewers for their encouragement and their input that helped us to further improve our review.

Round 2

Reviewer 2 Report

My last remark concerning this publication is: the column "Mechanical stimulus" in the Table1 rename to "Stimulus" , since there are various size and surface effects that have nothing to do with the mechanical stimulus.

The review has been improved and can be published.